# The Dominance Hierarchy of Wood-Eating Termites from China

**DOI:** 10.3390/insects10070210

**Published:** 2019-07-18

**Authors:** Theodore A. Evans, Boris Dodji Kasseney

**Affiliations:** 1School of Biological Sciences, University of Western Australia, Perth WA 6009, Australia; 2Laboratory of Applied Entomology, Faculty of Sciences, University of Lomé, BP 1515 Lomé 01, Togo

**Keywords:** *Macrotermes*, *Odontotermes*, *Coptotermes*, *Reticulitermes*, aggression tests, competitive release, urban ecosystems

## Abstract

Competition is a fundamental process in ecology and helps to determine dominance hierarchies. Competition and dominance hierarchies have been little investigated in wood-eating termites, despite the necessary traits of similar resources, and showing spatial and temporal overlap. Competition and dominance between five species of wood-eating termites found in Huangzhou, China, was investigated in three laboratory experiments of aggression and detection, plus a year-long field survey of termite foraging activity. Dominance depended on body size, with largest species winning overwhelmingly in paired contests with equal numbers of individuals, although the advantage was reduced in paired competitions with equal biomass. The termites could detect different species from used filter papers, as larger species searched through paper used by smaller species, and smaller species avoided papers used by larger species. The largest species maintained activity all year, but in low abundance, whereas the second largest species increased activity in summer, and the smallest species increased their activity in winter. The termite species displayed a dominance hierarchy based on fighting ability, with a temporal change in foraging to avoid larger, more dominant species. The low abundance of the largest species, here *Macrotermes barneyi*, may be a function of human disturbance, which allows subordinate species to increase. Thus, competitive release may explain the increase in abundance of pest species, such as *Coptotermes formosanus*, in highly modified areas, such as urban systems.

## 1. Introduction

A fundamental process in ecology is competition for resources between organisms with similar requirements [1,2,3]. Competition may be direct or indirect, for food, shelter, and other resources. Direct competition for food leads to dominance hierarchies, with the species able to use superior strength to dominate the most preferred food resource (e.g., for the classic experiments using granivorous insects in stored products, see [4,5,6,7]). The subordinate species may then specialise on less preferred food or access food in different locations or at different times than the dominant species [8,9,10,11].

Wood-eating termites offer an excellent system to consider competition and dominance hierarchies because the conditions for competition are present. Many species are found in the same habitat, or even in the same piece of wood [12], from the tropics to the warm temperate latitudes (ca. 35°) [13]. These termites utilise the cellulose in wood as food [14,15]. Studies on natural durability of wood against termites highlight wood species with high resistance, but they also show that termites do consume a wide range of wood species [16,17,18]. 

Although there are few studies that have considered whether competition and dominance hierarchies have evolved in termite communities, many studies have noted or used aggression in termites. These studies have mostly focussed on identifying colonies using aggression tests, beginning with [19], reviewed in [20,21], and/or identifying chemicals used to identify colonies [22,23,24,25]. Other studies have inferred competition between termite colonies due to the overdispersed spatial patterns of mound-nests [26,27,28]. Such studies strongly suggest that competition is occurring, but they do not attempt to measure its effects between species or infer competitive hierarchies. 

The aim of this study was to test for competition between five species of sympatric wood-eating termites in China, to identify a dominance hierarchy, and infer factors influencing pest status. The location in China had a moderately diverse generic diversity [13,29], including several species of fungus-growing Macrotermitinae, which appear to be the dominant wood-feeders when present, and some species of *Coptotermes* and *Reticulitermes* in the Rhinotermitidae, which often become pests in disturbed habitats [30,31,32]. The study system is therefore diverse enough to consider general patterns that may be relevant elsewhere in Asia and Africa, where fungus-growing termites are present, but not so diverse for the logistics of the experiment to be overwhelming.

## 2. Materials and Methods 

Termites were collected from the Hangszhou Botanic Gardens west of the West Lake, in the city of Hangzhou, Zheijiang province, China. The Botanic Gardens are contiguous with other gardens adjacent to the West Lake to the east, and with forest to the west, with a total wooded or forested areas of ca. 15 square kilometres. The termites were collected from dead wood lying on the ground or from wood-filled bait stations between October and November 2009.

The species were collected and identified [29] as (in decreasing body sizes): *Macrotermes barneyi* Light, *Odontotermes formosanus* (Shiraki), *Coptotermes formosanus* Shiraki, *Odontotermes hainanensis* Light, and *Reticulitermes flaviceps* (Oshima). Ten individuals from each caste of each colony of each species were weighed to determine average sizes (Table 1). Termites were separated from the wood in the laboratory and placed in ‘holding containers’, which were glass Petri dishes (diameter 200 mm) filled with rolled filter paper (rolls were 45 mm long x 6 mm diameter of Whattman No. 1 filter paper). The rolled filter paper gave the termites retreats in which to rest and receive food and moisture. The rolls were used in Experiment 3 (see below).

### 2.1. General Experimental Method

General setup for all experiments was as follows. Glass Petri dishes (diameter 90 mm) lined with moist filter paper were used for all experiments. Only workers were used in experiments, because soldiers were rare (sometimes less than one percent of individuals) and insufficient for appropriate replication. Similar petri dish experiments have been used in many studies [23,33,34,35,36], which have found that worker termites are usually aggressive against other termite species (for reviews see [21,37]). No individual was used twice. Termites used in the experiments were conducted between four hours and five days of collection, as aggression decreases with time in the laboratory [33,38].

### 2.2. Experiment 1: Equal Number of Termites

Five healthy (intact, walking normally) workers were chosen and placed into vertically placed Perspex tubes (∅15 mm) in the Petri dishes. The Perspex tubes were used to contain the termites and allow time for them to recover from relocation. In the experimental treatments, five workers of each of the two tested species were placed in two separate Perspex tubes. In the controls, five workers from the same colony were placed in two Perspex tubes. About five minutes after placement, when the termites had either slowed or stopped walking, the Perspex tubes were removed and the Petri dishes were covered. Petri dishes were observed for ca. 10 minutes and then placed in a constant temperature (25 °C) chamber for ca. 24 hours (22–26 hours), after which the numbers of live and dead termites were counted. Dead termites were separated into mutilated (presumed killed during fighting) and intact (presumed to have died for other reasons). Note that statistical analyses were performed on number of live termites only. There were five replicates for controls for all species, and 10 replicate dishes for each species combination, except for those with *O. hainanensis*, which had five replicates (due to low numbers of termites collected). 

### 2.3. Experiment 2: Equal Biomass of Termites

*Macrotermes barneyi* was chosen as the reference species for the equal biomass experiment, because it was the largest species. The reference biomass (42.1 mg) was based on five healthy (intact, walking normally) *M. barneyi* workers—three large workers plus two small workers, which was approximately the ratio found in field collections. The equivalent biomass of the other species comprised 8–10 workers of *O. formosanus*, 10–12 workers of *C. formosanus*, 22–24 workers of *O. hainanensis*, and 19–23 workers of *R. flaviceps* (Table 1). Note the variation in number of workers per replicate was due to variation in weight of individual termites, as the termites for each replicate were counted and weighed until they reached the reference biomass. The same protocol and same number of replicates were followed as for experiment 1.

### 2.4. Experiment 3: Chemical Detection of Species

This experiment tested whether termites could detect other species indirectly (chemically), from their habitations. The rolled filter papers (45 mm long × 6 mm diameter) from the ‘holding container’ Petri dishes were used. The termites had lived in the rolled filter papers (hence ‘rolls’) for up to seven days, and rolls with signs of inhabitation (chewing, building, and faecal staining) were chosen. Three rolls were placed on the inside edge in each experimental Petri dish at 120 ° to each other (forming a triangle): one roll from the same species (but different colony), one roll from another species, and one blank (unused) roll. Ten workers were chosen and placed into a Perspex tube in the centre of each Petri dish. About five minutes after placement, when the termites had stopped walking, the Perspex tubes were removed and the Petri dishes were covered. Petri dishes were placed in a CT chamber for 2 hours, then examined, then for a further 22 hours, after which the number and location of alive and dead termites were recorded in the same species roll, different species roll, blank roll, or not inside a roll. *Odontotermes hainanensis* was not used in this experiment, as there were too few individuals remaining from previous experiments. 

### 2.5. Field Survey

Species competing for the same resources may reduce competitive interactions spatially or temporally. To test this possibility, the dead wood on the ground in the Botanic Gardens was surveyed over one year for patterns of the termite species collected for the laboratory experiments. There were limitations, due to the management of the gardens, so attack on trees and some areas could not be surveyed. Therefore, a haphazard survey was designed. Four areas were designated, and transects were placed. The transects were walked, and the first 200 pieces of dead wood on the ground were examined for termite activity. The species identity was recorded when live termites were found.

### 2.6. Analysis

In all experiments, the numbers of live termites were converted to proportion survival in order to avoid issues with the different numbers of termites for each species used in experiment 2. For Experiments 1 and 2, the proportion survival for each species was compared against each pair combination using one-way ANOVA tests, with Bonferroni corrected post-hoc tests. For Experiment 3, the proportion of termites in or on the rolls of the same species, other species, or blank rolls were compared across species, after 2 and 24 hours, in a repeated measures ANOVA, and then within each species with an ANOVA after 2 hours. For the field survey, the number of pieces of dead wood with each termite species was compared over the 12 months of the year, and the proportion of dead wood occupied by termites was compared for each species by one-way ANOVA and MANOVA tests, with Bonferroni corrected post-hoc comparisons.

## 3. Results

### 3.1. Experiment 1: Equal Number of Termites

Termites in control treatments usually walked quickly around the petri dishes for five minutes or so, then slowed or stopped walking. During this time, they encountered other termites, usually stopping to antennate or without obvious change in behaviour. After one day, these termites had high survival (80–100%) and had usually chewed some of the filter paper in the petri dishes. Termites in mixed species treatments usually walked quickly around the petri dishes when first released, but when they encountered other heterospecific termites, they reacted either by lunging and biting (larger species) or retreating (smaller species). After 10 minutes, most individuals were either still walking rapidly or engaged in fighting. After one day, these termites had moderate (40–60%) to low (0–10%) survival, and there were no signs of chewed filter paper in the petri dishes. 

*Macrotermes barneyi* had a uniformly high level of survival regardless of co-placed species. Proportion survival averaged 0.80 ± 0.07 and did not differ across treatments (*F*_4,40_ = 1.770, *p* = 0.154; Figure 1A).

All other termite species had variable survival, which depended on co-placed species. Proportion survival of *O. formosanus* was significantly higher in controls and *R. flaviceps*, moderate to high with *O. hainanensis*, low with *C. formosanus*, and zero with *M. barneyi* (*F*_4,33_ = 33.178, *p* < 0.001; Figure 2A).

For *C. formosanus*, proportion survival was significantly higher in controls, with *O. hainanensis* and *R. flaviceps*, but significantly lower with *M. barneyi* and *O. formosanus* (*F*_4,35_ = 51.505, *p* < 0.001; Figure 3A).

Proportion survival of *O. hainanensis* was significantly higher in controls, medium with *R. flaviceps*, and significantly lower with the three other species (*F*_4,18_ = 16.207, *p* < 0.001; Figure 4A). For *Reticulitermes flaviceps*, the proportional survival was significantly higher in controls, with no difference with the four other species (*F*_4,35_ = 43.913, *p* < 0.001; Figure 5A).

*Macrotermes barneyi* had a more variable pattern of survival with equal biomass than for equal numbers. Proportion survival was significantly higher in controls and with *R. flaviceps*, significantly lower with *O. formosanus*, and moderate with all other species (*F*_4,34_ = 10.273, *p* < 0.001; Figure 1B). Proportion survival of *O. formosanus* was significantly higher in controls, moderate with *O. hainanensis* and *R. flaviceps*, and significantly lower with *M. barneyi* and *C. formosanus* (*F*_4,31_ = 32.442, *p* < 0.001; Figure 2B).

For *Coptotermes formosanus*, proportion survival was significantly higher in controls, moderate with *O. hainanensis* and *R. flaviceps*, and significantly lower with *M. barneyi* and *O. formosanus* (*F*_4,35_ = 30.126, *p* < 0.001; Figure 3B). Proportion survival of *O. hainanensis* was significantly higher in controls, medium with *R. flaviceps*, and significantly lower with the three other species (*F*_4,15_ = 48.738, *p* < 0.001; Figure 4B). For *Reticulitermes flaviceps*, the proportional survival was significantly higher in controls, with no difference with the four other species (*F*_4,35_ = 135.120, *p* < 0.001; Figure 5B). 

### 3.2. Experiment 2: Equal Biomass of Termites

The pattern of behaviours observed in ‘equal biomass’ were similar to those seen in ‘equal numbers of termites’ experiments: slow and calm movements in controls, fast and aggressive movements in mixed species. As the numbers of the smaller species of termite were higher, both the aggressive behaviour and mortality were higher.

### 3.3. Experiment 3: Chemical Detection of Species

After placement, termites walked around the petri dish, around, on, and through the rolled filter papers. The walking was variable; sometimes rapid, but rarely slow. After two hours, most of the termites had stopped walking and were motionless on one of the three filter papers, although some of the termites were still walking, albeit more slowly. After 24 hours, all termites had stopped walking and were inside rolled filter papers. The locations of termites varied from their own species and others, or blank papers, but did not change markedly over time (Figure 6). 

For *M. barneyi*, the majority (50–60%) of individuals were on their own paper when paired with paper from *O. formosanus*, but were on the other species paper when paired with *C. formosanus* and *R. flaviceps* (both 40–50%). Around 20–30% used blank paper. For *C. formosanus*, the majority of individuals (60%) were on blank paper when paired with paper from *M. barneyi*, but were mostly on the other species paper when paired with *O. formosanus* (50–75%) and *R. flaviceps* (50–60%). For *O. formosanus*, the majority of individuals were on their own paper when paired with paper from *M. barneyi* (55%) and *C. formosanus* (50–60%), as were a substantial minority when paired with paper from *R. flaviceps* (40%). For *R. flaviceps*, the majority of individuals were always on their own paper, whether paired with *M. barneyi* (75–85%), *C. formosanus* (45–60%), or *O. formosanus* (50–60%).

The repeated measures ANOVA found there was no effect of species (*F*_3,168_ = 0.004, *p* = 1.000), but a significant effect of paper source (own species, other species, or blank; *F*_2,168_ = 6.420, *p* = 0.002), but there was an interaction effect between them (*F*_6,168_ = 7.621, *p* <0.001). There was no effect of time, either alone (*F*_1,168_ = 0.021, *p* = 0.886), or as an interaction with species (*F*_3,168_ = 0.008, *p* = 0.999), paper source (*F*_2,168_ = 1.552, *p* = 0.215), or both species and paper source (*F*_6,168_ = 1.624, *p* = 0.143). Due to the significant interaction between species and paper source, and lack of effect of tie, each species was compared alone at 2 hours. 

For *M. barneyi*, there was no effect of species (*F*_2,36_ = 0.033, *p* = 0.967) or paper source (*F*_2,36_ = 1.433, *p* = 0.252), but there was an interaction effect between them (*F*_4,36_ = 9.258, *p* < 0.001). This was because there was no difference between number of *M. barneyi* on each paper source for *C. formosanus* and *R. flaviceps*, but they avoided *O. formosanus* paper in preference for their own. 

For *O. formosanus*, there was no effect of species (*F*_2,36_ = 0.031, *p* = 0.969), but there was a significant effect of paper source (*F*_2,36_ = 8.917, *p* = 0.001), and there was also an interaction effect between them (*F*_4,36_ = 2.890, *p* = 0.036). This was because *O. formosanus* generally preferred their own paper and other species paper to blank paper, except for paper from *M. barneyi*, which they avoided. 

For *C. formosanus*, there was no effect of species (*F*_2,36_ = 0.000, *p* = 1.000) or paper source (*F*_2,36_ = 2.528, *p* = 0.094), and there was no interaction effect between them either (*F*_4,36_ = 1.056, *p* = 0.392). This was because *C. formosanus* had a relatively variable and thus not significant response, though the general pattern was they preferred blank paper to their own paper and that from *M. barneyi*, and they preferred other species paper from *O. formosanus* and *R. flaviceps* to either their own or blank paper. 

For *R. flaviceps*, there was no effect of species (*F*_2,36_ = 0.002, *p* = 0.998), but there was a significant effect of paper source (*F*_2,36_ = 7.551, *p* = 0.002), and there was no interaction effect between them (*F*_4,36_ = 1.331, *p* = 0.277). This was because *R. flaviceps* preferred their own paper to and other species paper and blank paper, in all circumstances. 

### 3.4. Field Survey

There was a clear pattern of more termite activity in the warmer months of the year, with over 40% of the dead wood containing termites from June through to October, with a peak in September (Figure 7). The cooler months of the year had the lowest termite activity, lower than 10% of dead wood with termites.

The majority of the termites found in the dead wood were *O. formosanus*, then *O. hainanenis*, *R. flaviceps*, *M. barneyi*, and, rarely, *C. formosanus*. There were very occasional appearances of other species (*Pericapritermes jangtsekiangensis* and *Sinotermes mushao*). The number of pieces of wood with termites did vary over the year for *O. formosanus* (1–45/month; *F*_11,36_ = 23.459, *p* < 0.001), but did not vary over the year for *M. barneyi* (1–5/month; *F*_11,36_ = 0.727, *p* = 0.706), *C. formosanus* (0–1/month; *F*_11,36_ = 1.987, *p* = 0.060), *O. hainanensis* (1–12/month; *F*_11,36_ = 1.772, *p* = 0.096), or *R. flaviceps* (1–6/month; *F*_11,36_ = 1.241, *p* = 0.297 (Figure 7). 

Considering only the wood in which termites were found, *O. formosanus* was the most common, with an average proportion across the year of 0.59 of all termites (ranging from 0.23 to 0.83). *O. hainanensis* averaged 0.13 (0.00–0.38), *R. flaviceps* averaged 0.19 (0.01–0.67), *M. barneyi* averaged 0.07 (0.00–0.19), and *C. formosanus* averaged 0.001 (0.000–0.006) (Figure 8).

The ratio of these proportions could change, sometimes dramatically, over the year. For *O. formosanus*, there were significantly higher proportions in attacked wood pieces in summer (*F*_11,35_ = 6.049; *p* < 0.001), whereas for *O. hainanensis*, there were a significantly higher proportions in early winter, and for *R. flaviceps*, there were a significantly higher proportions in late winter (*F*_11,35_ = 4.513; *p* < 0.001). There were no changes in proportions of attacked wood over the year for either *M. barneyi* (*F*_11,35_ = 1.297; *p* = 0.267), or *C. formosanus* (*F*_11,35_ = 0.832; *p* = 0.610).

## 4. Discussion

Termites displayed high levels of aggression, typically fighting to the death. Body size was the most important factor affecting survival, regardless of whether equal numbers or biomass of termites were used. The advantage of a large body size difference was overwhelming for equal numbers, and lessened but still important for equal biomass. Termites detect the presence of their own or other termite species from the papers, and their positioning matched the results of the aggression experiments. In general, each species avoided the papers of larger species, and explored the papers of smaller species. Termite foraging activity was generally higher in the summer months, but this varied between species. The largest species, *M. barneyi*, did not change foraging over the year, the most abundant species, *O. formosanus*, increased foraging activity dramatically in summer, and the two smaller species, *O. hainanensis* and *R. flaviceps*, were relatively more abundant in winter.

Taken together, this study suggests that these wood-eating termite species are competing, have evolved a dominance hierarchy, and subordinate species actively avoid dominant species. The dominant species was *M. barneyi*, with *O. formosanus* the first subordinate species, perhaps subdominant, followed by *C. formosanus*, then *O. hainanensis*, and finally *R. flaviceps*. In the Philippines, *Macrotermes gilvus* was found to attack and kill *Coptotermes gestroi*, *Nasutitermes luzonicus*, and *Microcerotermes losbanosensis* in similar petri dish experiments [35]. The smaller species may have changed their foraging activity to avoid the more dominant larger species [8,9,10,11]. Although this is the first study that proposes a dominance hierarchy based on competition for termites, they have been proposed for other social insects, including ants [39,40,41] and bees [42,43,44,45].

The hierarchy is based mostly on body size, as with other animals [8], but may be ameliorated by biomass of fighting termites. In any encounter, the biomass of potential fighters would be based in part on the colony size. The population sizes of these species in China are not known; however, the colonies of *Macrotermes* species in Africa are very populous (several millions), those of *Coptotermes* species in Australia and Asia somewhat less (half to one million), and those of *Reticulitermes* species in the USA and Japan much smaller again (tens to hundreds of thousands)—for reviews, see [46,47]. If the population sizes of colonies of the Chinese species match those of their congenerics elsewhere, it would also match the dominance hierarchy observed in the present study. 

Termites recognised the habitations (the filter papers) of other species and either avoided those that were more dominant or searched through those that were subordinate. The recognition was likely chemical, either from pheromones secreted by the termites, perhaps deliberately, cuticular hydrocarbons that had rubbed onto paper from movement, or from faeces deposited on the paper—or a combination of all three. All three of these potential signals differ between termites—e.g., pheromones [48,49,50,51,52], gut bacteria [14,15]—but the exact mechanism is unknown. Given that there are termite species that are inquilines of other termite species, such chemical signals may be attractive (to find hosts) as well as repellent (to avoid more dominant competitors) [53]. Note that termites use vibrational signals to find conspecifics [54], competitors [12], and predators such as ants [55], and these signals may be of importance when the termites themselves are active.

The role of competition is not likely to be the only explanation for community structure. Other abiotic and biotic factors are important as well, including weather events, disturbance, microhabitat specialisation, predators, and parasites—e.g., for ants [56,57,58] and for bees [42]. This is likely to be so for termites as well and may explain some of the patterns found in the current study. The lab data suggested *Macrotermes* would be dominant, but the field data showed it was less common than either of the two *Odonotermes* species. This may be because the sampling did not include trees or large fallen branches, and the size of food resources does influence foraging decisions of species of *Macrotermes*, *Coptotermes* and *Reticulitermes* [59,60,61], and other species [62,63,64]. The lack of sampling of large food resources was an unintended artefact of human disturbance (management) in the Botanical Gardens.

Human disturbance is well known to change termite communities in all biogeographic provinces [65,66,67,68,69]. The most sensitive species belong to the soil-feeding and soil-interface functional groups. The fungus-growing termites are also sensitive to disturbance, with larger bodied species more likely to disappear (West Africa [66], SE Asia [67,70,71,72]). Therefore, *M. barneyi* may have been less abundant in the human managed botanical gardens, than in completely undisturbed native forests. In West Africa, another dominant *Macrotermes* species, *M. bellicosus*, was found to be important in structuring the rich termite communities in undisturbed primary forest, but not so in species poor disturbed agricultural land [73].

Species in the Rhinotermitae, such as *Coptotermes*, become more abundant with disturbance: in logged natural forests [66,67]; in plantations [30,31,74,75,76,77]; and particularly in urban areas [78,79]. The main *Coptotermes* species in China, *C. formosanus*, was encountered rarely in the survey of the partially disturbed Botantic Gardens field site, but just a few hundred meters away it is a major pest in urban areas, as is common in China [80].

*Coptotermes* is believed to be rare in forests, because it is rarely encountered in standard transect surveys [65,66,67,68,69]. However, this result may be a sampling artefact, as these surveys do not sample within standing, live tree trunks, which is where *Coptotermes* evolved to live [12,81]. *Coptotermes* may have evolved this habit in order to avoid competition with other termites, such as *Macrotermes* and other fungus-growing species. Although *Coptotermes* is from the Rhinotermitidae, which is a more basal family than the Macrotermitidae, the genus is younger, having evolved about 25 million years later than the Macrotermitidae [82,83]. It is possible that *Coptotermes* is more common in forests, hiding inside tree trunks, than typically assessed.

Either way, *Coptotermes* may still increase in abundance from rare(r) in undisturbed forest to common in urban areas. This may be driven due to the loss of the more dominant fungus-growing termites from human modification of the landscape: ‘competitive’ (or ‘ecological’) release [84]. It may be that human disturbance actually selects for pest species, perhaps with r-selected characteristics [85]. The same may be true for *Reticulitermes* in cooler locations, such as Japan. It is also possible that *Coptotermes* has a greater tolerance to the higher temperatures found in urban areas [86]. 

Therefore, an increase in *Coptotermes* due to disturbance may be observed due to the absence of the more dominant fungus-growing termites. Fungus-growing termites did not disperse to Australia or New Guinea, whereas *Coptotermes* did, arriving from Asia around 15 Ma and 5 Ma respectively [82], likely before the Termitidae [83]. *Coptotermes* diversified and dominate Australian forests, with three species evolving mound building habits [81]. Several *Coptotermes* species have been introduced by humans to areas without fungus-growing termites and have since become abundant, including in forests in Japan, Madagascar, and the USA [87,88]. 

## 5. Conclusions

This study found strong aggression competition between five species of wood-eating termites in China. Body size was the most important factor in competition, but this was reduced when biomass was equivalent. The termites detected and responded to used papers (likely chemical signals), with larger species searching through papers from smaller species, and smaller species avoiding papers used by larger species. Finally, activity in the field suggested there was temporal (seasonal) avoidance, with smaller species more active in the winter months. These results suggest there is a dominance hierarchy among the termites in China. 

It is likely that dominance hierarchies will vary between termite communities in different biogeographic areas. Africa has the highest diversity of termites, including fungus-growing species [13], so it seems likely that fungus-growing termites will dominant the hierarchy in Africa, as observed in the current study in China. However, Africa has diverse wood-eating termites in the Nasutitermitidae and Amitermitinae [13,65,66], which were not found in the current study, and so their rank in a hierarchy remains unknown. South America is likely to be different again, as there are no fungus-growing termites, but many Nasutitermitidae and Syntermitinae [13,68,69]. Australia is dominated by *Coptotermes* to a greater degree than any other biogeographic zone, including the only mound-building species [81], suggesting a different pattern of dominance again. There remains much research to uncover competitiveness and dominance hierarchies across all biogeographic zones, and common patterns, if any, can be ascertained.

## Figures and Tables

**Figure 1 insects-10-00210-f001:**
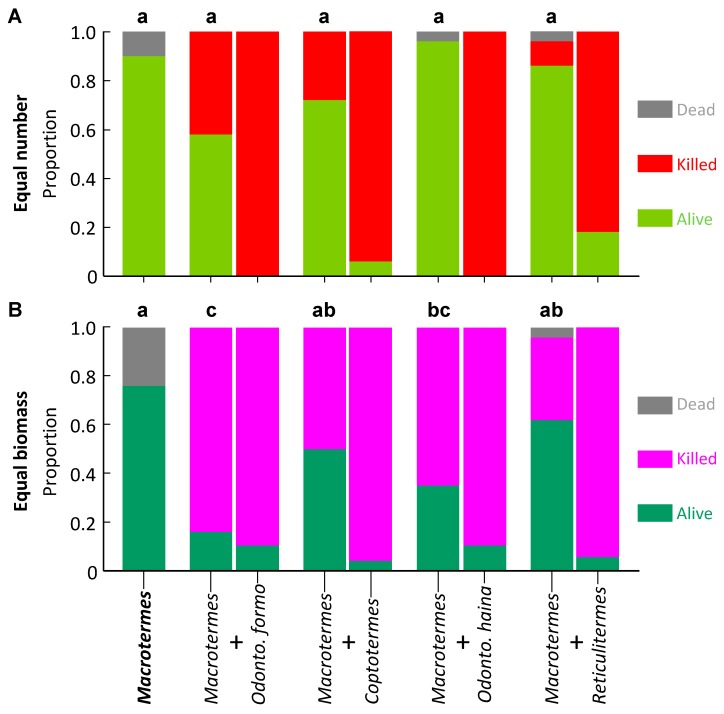
Survival of *Macrotermes barneyi* in laboratory aggression experiments. The control appears as a single column as it contains one species, whereas paired species treatments are paired columns, the *M. barneyi* on the left of each pair. Columns surmounted by different letters signify significantly different proportion survival (Bonferroni corrected pairwise comparisons) for *M. barneyi*. (**A**) Equal number of termites; (**B**) Equal biomass of termites.

**Figure 2 insects-10-00210-f002:**
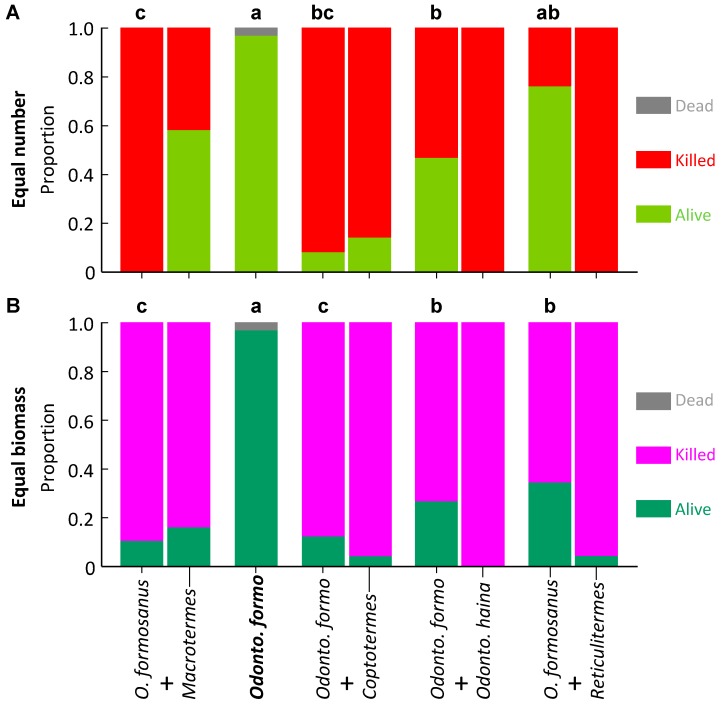
Survival of *Odontotermes formosanus* in laboratory aggression experiments. The control appears as a single column as it contains one species, whereas paired species treatments are paired columns, the *O. formosanus* on the left of each pair. Columns surmounted by different letters signify significantly different proportion survival (Bonferroni corrected pairwise comparisons) for *O. formosanus*. (**A**) Equal number of termites; (**B**) Equal biomass of termites.

**Figure 3 insects-10-00210-f003:**
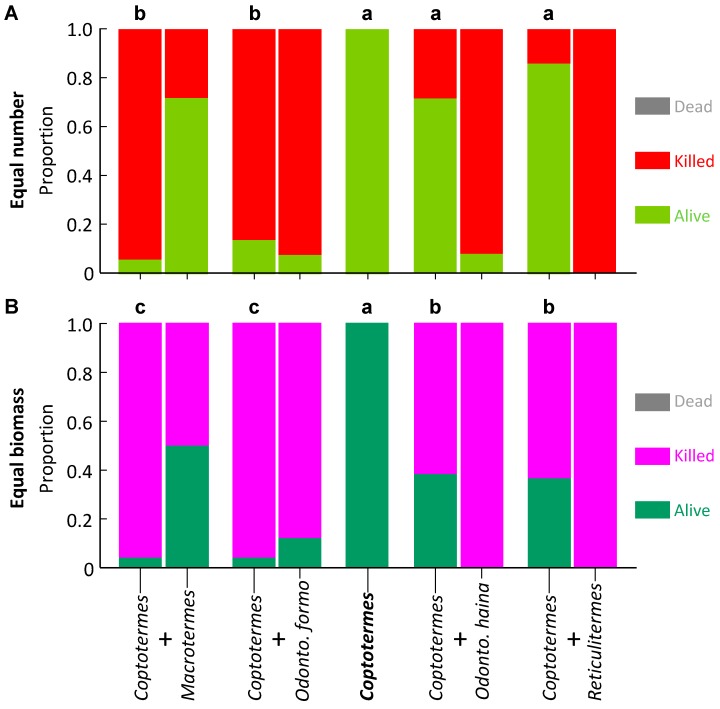
Survival of *Coptotermes formosanus* in laboratory aggression experiments. The control appears as a single column as it contains one species, whereas paired species treatments are paired columns, the *C. formosanus* on the left of each pair. Columns surmounted by different letters signify significantly different proportion survival (Bonferroni corrected pairwise comparisons) for *C. formosanus*. (**A**) Equal number of termites; (**B**) Equal biomass of termites.

**Figure 4 insects-10-00210-f004:**
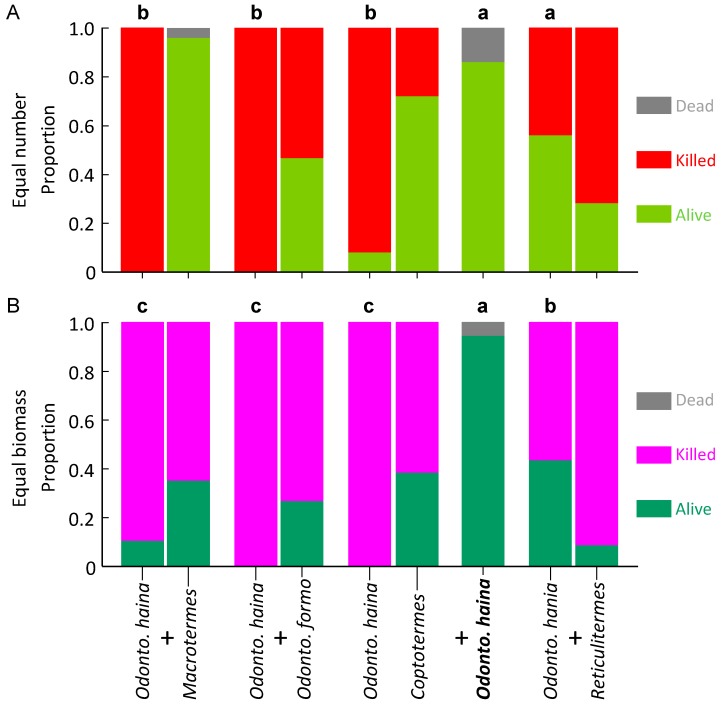
Survival of *Odontotermes hainanensis* in laboratory aggression experiments. The control appears as a single column as it contains one species, whereas paired species treatments are paired columns, the *O. hainanensis* on the left of each pair. Columns surmounted by different letters signify significantly different proportion survival (Bonferroni corrected pairwise comparisons) for *O. hainanensis*. (**A**) Equal number of termites; (**B**) Equal biomass of termites.

**Figure 5 insects-10-00210-f005:**
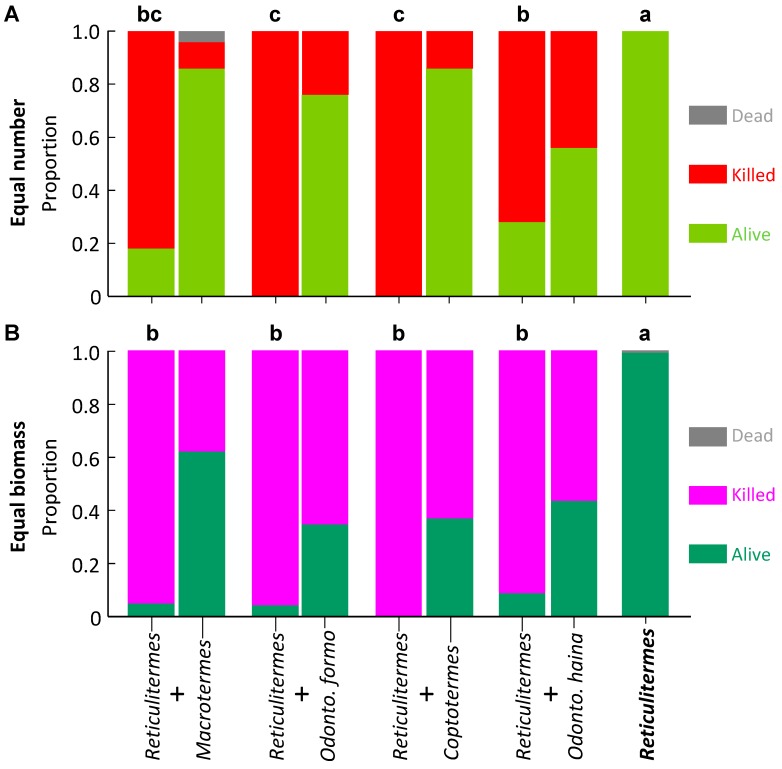
Survival of *Reticulitermes flaviceps* in laboratory aggression experiments. The control appears as a single column as it contains one species, whereas paired species treatments are paired columns, the *R. flaviceps* on the left of each pair. Columns surmounted by different letters signify significantly different proportion survival (Bonferroni corrected pairwise comparisons) for *R. flaviceps*. (**A**) Equal number of termites; (**B**) Equal biomass of termites.

**Figure 6 insects-10-00210-f006:**
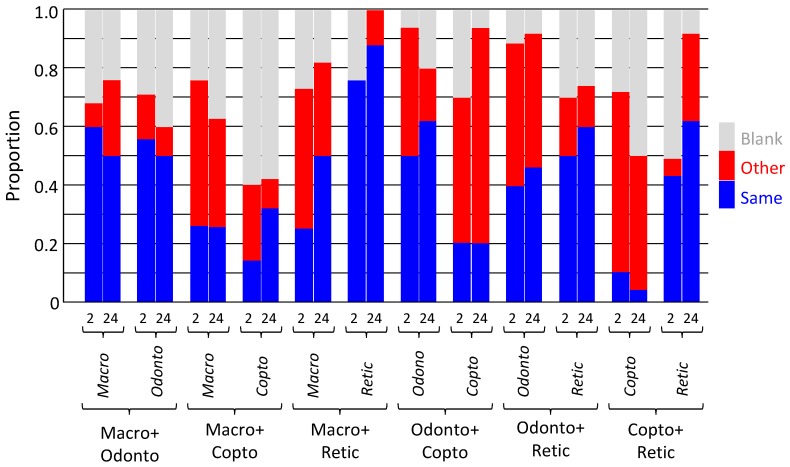
Location of termites in chemical detection experiment. Termites were offered three pieces of rolled filter paper: from the same species (another colony), other species, or blank. Locations recorded after 2 and 24 hours. Macro = *Macrotermes barneyi*; Odonto = *Odontotermes formosanus*; Copto = *Coptotermes formosanus*; Retic = *Reticulitermes flaviceps*.

**Figure 7 insects-10-00210-f007:**
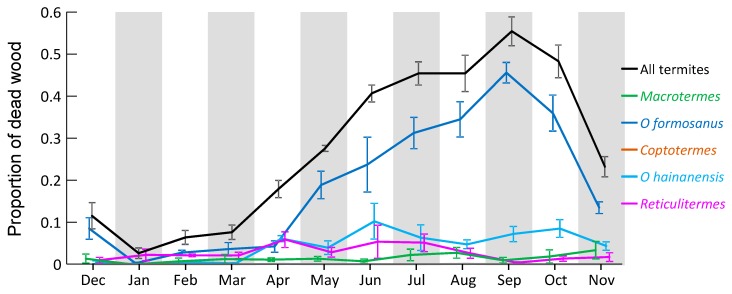
Termite activity over one year in Hangzhou Botanical Gardens, China. Data are average (± standard error) proportion of dead wood on ground containing termites. There were 200 pieces of wood sampled each month.

**Figure 8 insects-10-00210-f008:**
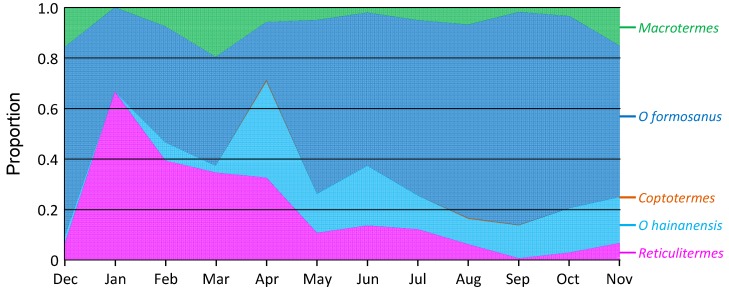
Proportional activity of various termite species in wood containing termites, over one year in Hangzhou Botanical Gardens, China. Data are average (± standard error) proportion of dead wood on ground containing each termite species.

**Table 1 insects-10-00210-t001:** The weights (mean ± standard error) of the different castes for each termite species. Note *M. barneyi* has two castes of workers (large and small). Only workers were used in the laboratory experiments; soldier weights are provided for comparison only. Numbers of individuals for equal biomass (mean ± standard error) were based on the average weight of 5 *M. barneyi* workers (3 large + 2 small) of 42.1 mg.

Species	Worker		Soldier	Number of Individuals for Equal Biomass
*Macrotermes barneyi*	large	10.41 ± 0.54	22.43 ± 0.52	3
	small	5.45 ± 0.32	12.01 ± 1.67	2
*Odontotermes formosanus*		4.99 ± 0.19	4.57 ± 0.13	8.8 ± 0.9
*Coptotermes formosanus*		4.05 ± 0.06	4.43 ± 0.06	10.9 ± 1.3
*Odontotermes hainanensis*		2.13 ± 0.06	2.11 ± 0.05	23.0 ± 1.0
*Reticulitermes flaviceps*		2.08 ± 0.05	2.52 ± 0.05	21.1 ± 2.2

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
