# Peer review of "The Dominance Hierarchy of Wood-Eating Termites from China"

_insects, 2019, doi:10.3390/insects10070210_

Round 1

Reviewer 1 Report

See attached pdf.

Reviewer 2 Report

See attached PDF for comments and suggested changes.

Interesting study.

Ln 11-16 Good abstract, but not sure about leaving the 1) out for background.  Maybe remove the 2) and 3) numbers and just have Background, Methods and Results sentence heading.  Or leave it.

Ln 68-69 Please add authority for species

Ln 79-83 Change margin on table title. 

Ln 154-160 This paragraph has many unnecessary commas

Ln 333-336 Missing reference 54

ln 440 fix indentation on reference 18

Reviewer 3 Report

This is a well executed applied ecological study that presents information important to understanding the termite community present in the specific location.  It is valuable information for any future pest control measures that need to be taken. 

I find one fundamental missing aspect of this study--how were termites identified to species?  What keys were used? Of the species studied, which are introduced and which are native to the region.  I'd like to see a paragraph added in Methods discussing how species were identified, since Reticulitermes as a genus still has unresolved cryptic species. I'd also like to see a paragraph in the Discussion added about introduced versus native species and how that can contribute to competition. 

Line 68: Please justify species identification methods here. What key or keys were used? 

Line 102: What was the constant temp? List. 

Lines 62-149: I recommend considering an additional Protocol paper with more detailed pictures of the rolls and the methods.  It would enhance the paper and improve reproducibility.  Consider Bio-Protocol (will go through an additional editing and peer review--so will count as another publication) or Protocol.io. This is more utilized by molecular researchers, but it would be highly beneficial for behavioral work as well, particularly because behavioral bioassays can vary highly between researchers. For the chemical detection of species, instead of adding pictures to this paper, writing a Protocol paper with images would be valuable.  I marked "rolled filter papers" as needing pictures in lines 120-121 and lines 223-224.  If non Protocol paper is utilized, then some simple pictures or graphics of the bioassays would be helpful. 

Lines 307-308  In writing about an "evolved dominance hierarchy", here is where a couple sentences or a paragraph could be added about introduced species. Isn't R. flavipes a non-native species in China? If yes, then introduced species have colony changes that can also enhance their competitive abilities (for example, the Argentine ant in the USA or C. formosanus in Hawaii and the American South). Such colony structure changes result in a loss of nestmate recognition and huge colony sizes. We can't talk about anything "evolved" without talking about what species are native and what ones are not. A paragraph in the discussion about introduced species and colony changes and competitive advantages is warranted here. You should speak to the evolutionary history of the species. 
